# Time-Restricted Fasting Improves Liver Steatosis in Non-Alcoholic Fatty Liver Disease—A Single Blinded Crossover Trial

**DOI:** 10.3390/nu15234870

**Published:** 2023-11-22

**Authors:** Jack Feehan, Alexandra Mack, Caroline Tuck, Jorge Tchongue, Darcy Q. Holt, William Sievert, Gregory T. Moore, Barbora de Courten, Alexander Hodge

**Affiliations:** 1Institute for Health and Sport, Victoria University, Melbourne 3011, Australia; jack.feehan@vu.edu.au; 2School of Clinical Sciences, Monash University, Melbourne 3168, Australia; alliemack333@gmail.com (A.M.); william.sievert@monash.edu (W.S.); gregory.moore@monash.edu (G.T.M.); 3Department of Nursing and Allied Health, Swinburne University of Technology, Hawthorn 3122, Australia; 4Gastroenterology and Hepatology Unit, Monash Health, Melbourne 3168, Australiadarcy.holt@monash.edu (D.Q.H.); 5School of Health and Biomedical Sciences, RMIT, Bundoora 3083, Australia

**Keywords:** non-alcoholic fatty liver disease, metabolic syndrome, intermittent fasting, time-restricted eating, diet, fatty liver, obesity

## Abstract

Background: Non-alcoholic fatty liver disease (NAFLD) is associated with visceral adiposity. We assessed the effectiveness of time-restricted fasting (TRF) for 16 h daily without calorie restrictions compared to standard care (SC; diet and lifestyle advice) in improving visceral adiposity and steatosis via controlled attenuation parameter (CAP). Methods: In a prospective single-blind randomized controlled trial, 32 participants with NAFLD were randomly assigned to TRF or SC for 12 weeks. The secondary endpoints were changes in liver stiffness, anthropometry, blood pressure, and other metabolic factors. Results: Twenty-eight participants completed the first arm of the study (TRF = 14, SC = 14), with 23 completing the crossover arm (TRF = 10, SC = 13). The baseline demographics were similar between the groups. Intermittent fasting caused a significant decrease in hepatic steatosis (*p* = 0.038), weight (*p* = 0.005), waist circumference (*p* = 0.001), and BMI (*p* = 0.005) compared to standard care. Intermittent fasting also resulted in additional within-group changes that were not seen in the standard care intervention. Conclusion: TRF offers superior improvements in patients with NAFLD, improving steatosis, weight, and waist circumference despite a lack of change in overall caloric intake. Time-restricted fasting should be considered as a primary weight loss intervention in the context of NAFLD. Trial registration: ACTRN12613000935730.

## 1. Introduction

Non-alcoholic fatty liver disease (NAFLD) is the most common chronic liver disease worldwide, with a global prevalence of 29.8% [1]. NAFLD and metabolic syndrome are intrinsically and bi-directionally linked [2]—a constellation of central obesity, impaired glucose tolerance, dyslipidaemia, and hypertension [3]. NAFLD patients with metabolic syndrome have an increase in all-cause, liver–specific, and cardiovascular mortality [4,5].

Due to the high prevalence of NAFLD and its associated liver and non-liver complications, there is a pressing need for effective treatment strategies. As NAFLD is inextricably linked to obesity and metabolic syndrome through over-nutrition [6], treatment has focused on forms of calorie restriction and adherence to the Mediterranean diet [7]. Prolonged periods of fasting or a calorie-restricted diet results in weight loss and may improve liver steatosis [8]. Reducing caloric intake also has positive effects, improving glucose tolerance and insulin sensitivity and hypertension and increasing lifespans [9].

CR diets are often difficult to incorporate into the lifestyles of typical patients; therefore, adherence is often poor and inversely proportional to the severity of caloric restriction [10]. An alternative approach is to alter the timing of energy intake through reduced meal frequency or fasting. Time-restricted fasting (TRF) is a form of fasting in which individuals consume ad libitum within a window of time each day (commonly 8 h) and fast for the remainder of the day. An important point of difference between calorie restriction and TRF is that the overall caloric intake with TRF need not be reduced—only the frequency and timing of consumption are altered [9,11]. Consequently, TRF can be easier to understand and follow. While TRF has been routinely shown to induce effective weight loss [12], its effect on hepatic health has not been studied. Therefore, the aim of this pilot study was to determine whether a 12-week period of TRF in individuals with NAFLD is an acceptable form of dietary modification and whether it had similar accessible point-of-care clinical and biochemical outcomes to individuals given standard care (SC) lifestyle advice regarding CR and exercise.

## 2. Materials and Methods

### 2.1. Study Design

This 12-week randomized, controlled, single-blind, AB/BA crossover pilot study was conducted at Monash Health, Melbourne, Australia. All participants were recruited from outpatient clinics and general practice and screened in our gastroenterology outpatient clinics. All provided signed informed consent. The study was approved by Monash Health Human Research Ethics Committee A (registration number: HREC 12298A, 12 January 2013) and registered at the Australian New Zealand Clinical Trials Registry (registration number: ACTRN12613000935730). Eligible subjects were between 18 and 75 years of age with an ultrasound diagnosis of fatty liver [13] and met inclusion/exclusion criteria (Appendix A). Subjects allocated to the TRF arm were instructed to refrain from consuming any and all food and energy-containing drinks for 16 h from 8 pm to 12 noon the following day. During this time, water, black tea, and black coffee were permitted. Between the hours of noon and 8 pm (8 h), subjects were able to consume food as desired, without any prescribed caloric restriction. Adherence to the fasting intervention was assessed via participant self-report, with days of non-compliance with fasting and the reasons for non-compliance being recorded. Those allocated to the SC arm were instructed to follow the exercise and CR advice provided by the Gastroenterological Society of Australia (GESA), which recommends a weight management program (aiming for 0.25–0.5 kg of weight reduction per week), along with a diet low in calories and fat, as well as mixed aerobic and resistance exercise 5 days per week. Participants were provided with the GESA information sheet on NAFLD for further references and information. This information was provided by a member of the investigator team at random. Participants in both groups were contacted by a member of the study team at fortnightly intervals throughout the study to facilitate compliance to dietary interventions.

The primary endpoints were a reduction in hepatic steatosis (determined via transient elastography, controlled attenuation parameter (CAP), and visceral adiposity over the course of the study). All co-authors had access to the study data and have reviewed and approved the final version of the manuscript.

### 2.2. Randomization and Masking

All participants were randomized 1:1 via blinded envelope to either the TRF or SC arm through the use of a randomization software using predetermined permuted blocks of six, and the randomization sequence was prepared by an independent statistician. Investigators collecting data and performing analyses were blind to the group allocation throughout the study. Due to the nature of the intervention, blinding of the participants was not possible.

### 2.3. Primary Outcomes

Liver steatosis was determined via CAP measurement by FibroScan^®^ (Echosens, Paris, France) with the M-probe transient elastography, which simultaneously assessed liver stiffness. All participants fasted overnight. Only LSM and CAP studies with ≥10 valid readings and a median measurement with an interquartile range/median value of ≤30% were used in the analysis. FibroScan^®^ studies were performed at 0, 12, and 24 weeks. The single operator was blinded to the participants’ study assignment. Visceral adiposity was determined via abdominal computerized tomography (CT). All CT scans were performed at Monash Medical Centre at 0, 12, and 24 weeks using a GE Lightspeed (General Electric Medical Systems, Milwaukee, WI, USA). A single abdominal slice was taken at the L4/5 level. The L4–5 level was analysed for visceral fat area (VFA), as a single slice at this level has been shown to correlate highly with total visceral fat volume [14]. Digital Imaging and Communications in Medicine images were imported and analysed for body composition using SliceOmatic 4.3 (TomoVision, Montreal, QC, Canada). Hounsfield unit (HU) ranges were used to differentiate between components of body composition on the images. Tissue from −30 to −190 HU was segmented as fat, and tissue from −30 to 150 HU was segmented as muscle. Fat inside the abdominal muscle layer was then differently segmented as visceral fat, and fat outside the abdominal muscle wall was considered as subcutaneous fat. Any fat inside muscle was segmented as inter-muscular fat. VFA was calculated from the sum of areas for the relevant segments. A single experienced observer (blinded to the participants grouping) performed all SliceOmatic segmentations. Overall, 12 out of 14 in the TRF group and 11 out of 14 in the SC group had images suitable for analysis at baseline and week 12.

### 2.4. Secondary Endpoints

#### 2.4.1. Anthropometry

Participants were weighed in hospital gowns using the same Wedderburn DS-530 Handrail scale, with weights recorded to the nearest 0.1 kg. Height was measured using a stadiometer and recorded to the nearest 0.5 cm. WC was measured at the midpoint between the superior aspect of the iliac crests and the inferior margin of the ribs, recorded to the nearest 0.5 cm. Anthropometry measurements were performed by an investigator blinded to the participants’ grouping. Body composition for total body fat and lean mass was assessed by a blinded radiographer at Monash Medical Centre using the same GE lunar Prodigy DEXA in standard mode (software version 13.60).

#### 2.4.2. Blood Pressure

This was performed by an investigator blinded to the participants’ grouping using a manual sphygmomanometer. With the patient seated, a single reading was taken on the right arm.

#### 2.4.3. Biochemical Assessments

All participants were asked to fast for at least nine hours prior to venepuncture. Monash Health Pathology, using assays accredited by the National Association of Testing Authorities, processed blood samples for ALT, insulin, glucose, total cholesterol, triglyceride and stored serum at −80 °C. The stored serum was used for TNF-α, adiponectin, and leptin analysis via ELISA according to the manufacturer’s instructions (R&D Systems, Minneapolis, MN, USA & Cusabio, Wuhan, China). Analysis was performed using a microplate reader at 490 nm for TNF-α and 450 nm for adiponectin and leptin (Magellan, Phoenix, AZ, USA). The TNF-α, adiponectin, and leptin concentrations were calculated from the standard curve generated in GraphPad Prism 5.0 d for Mac OS X (GraphPad Software, La Jolla, CA, USA). To measure faecal calprotectin, faecal material was collected by individuals at home and frozen at −20 °C before being delivered to the investigators. Samples were then stored at −80 °C until the end of the study. Faecal samples were processed using Buhlmann Smart-Prep Extraction kits, followed by Buhlmann Calprotectin ELISA, following the manufacturer’s instructions (Taylor Scientific, St Louis, MO, USA). Analysis was performed using a microplate reader at 450 nm (Magellan). Faecal calprotectin concentration was calculated from the standard curve generated in GraphPad Prism 5.0 d for Mac OS X (GraphPad Software, La Jolla, CA, USA).

#### 2.4.4. Dietary and Appetite Monitoring

Dietary changes were measured via a validated 3-day food diary self-reported at baseline and every two weeks. Total daily calorie, fat, carbohydrate, and protein intake were calculated using Foodworks Professional (Version 7, Xyris Software, Kenmore, Queensland, Australia). The 3-day average was used to estimate daily intake. Questionnaires were used throughout the study to determine the impact of TRF and SC on hunger, activity, and quality of life. Hunger, satisfaction, fullness, and happiness with diet were assessed using a visual analogue scale [15].

#### 2.4.5. Global Physical Activity Questionnaire (GPAQ)

To identify any changes in exercise which may have influenced outcomes over the course of the trial, participants were asked to complete the GPAQ at baseline, week 12, and week 24. The GPAQ consists of 19 questions, assessing the time spent per week partaking in physical activity across three domains: work, transport, and leisure. Within these domains, physical activity was further categorized into two sub-domains: vigorous or moderate activity. Based on the GPAQ responses, the number of Metabolic Equivalent (MET-) minutes achieved by each patient per week were calculated. The calculation of MET-minutes and data cleaning were performed in accordance with the GPAQ Analysis Guide.

#### 2.4.6. Statistical Analysis

Sample size was calculated based on a validation study of FibroScan^®^ CAP measurements for assessing liver steatosis [16], with target recruitment set at 58 participants to give 80% power at 95% confidence; due to administrative changes at the study site, the trial was halted after 32 participants had been recruited. Distribution was evaluated via Shapiro–Wilk testing. A paired analysis (paired *t* test or Wilcoxon signed rank test) of within-individual change was used to identify the differences between the interventions, with each individual acting as their own internal control for the analysis [17]. Mixed linear models were used to evaluate within-group changes over the two 12-week intervention periods. The balanced design of the crossover ensured that any period effects were likely evenly distributed across interventions. Missing data were random and managed via pairwise deletion. An alpha value of 0.95 was used, resulting in a *p* value < 0.05 being considered significant.

## 3. Results

### 3.1. Patient Flow

Thirty-two subjects were enrolled in the study; seventeen belonged to the TRF group, while fifteen belonged to the SC group (Figure 1). Three did not complete the trial, and one was lost to the follow-up period. A total of 14 subjects in each group were analysed at 12 weeks, with 10 (TRF) and 13 (SC) crossing over and being analysed at 24 weeks.

### 3.2. Baseline Characteristics

The demographic and anthropometric characteristics of each group were similar at baseline (Table 1).

### 3.3. Primary Outcomes

#### 3.3.1. Transient Elastography

The intermittent fasting intervention led to a decrease in CAP compared to standard care, with a mean decrease of 9.96 (±51.03) dB/m with TRF compared to an increase of 20.46 (±49.15) dB/m with SC (*p* = 0.038, Table 2). Participants in the TRF group showed significant decreases in CAP over the 12 weeks (*p* = 0.049), while those in the SC group showed an increase in CAP over the same period (*p* = 0.05, Table 3). A subgroup of those with CAP values > 268 dB/m (*n* = 10 [TRF], 10 [SC]), the cut-off considered for significant steatosis [18,19], demonstrated even larger decreases in CAP values after the TRF intervention, with a mean decrease of 26.10 (±33.01) dB/m compared to an increase of 15.10 (±17.49) dB/m with SC (*p* = 0.035, Table 4).

#### 3.3.2. Visceral Fat

There was no significant change in visceral fat between the two interventions (*p* = 0.702, Table 2). However, only the TRF intervention led to a decrease in visceral fat over the 12 weeks (*p* = 0.029, Table 3), with the SC intervention showing no significant changes.

#### 3.3.3. Secondary Outcomes

##### Hepatic Measures

There were no significant between-group or within-group differences in liver stiffness over the course of the trial. The TRF intervention alone led to a decrease in ALT concentrations over the course of the study (*p* = 0.005); however, while there was a trend toward a difference between interventions (*p* = 0.094), this did not reach predetermined statistical significance. There were no differences in either liver stiffness or ALT in the subgroup analysis of participants with significant steatosis.

##### Anthropometry, Lipid Profiles, and Blood Pressure

Compared to the SC intervention, the TRF intervention caused significantly greater decreases in weight ([TRF] −1.72 ± 2.01 vs. [SC] 0.13 ± 2.13, *p* = 0.005), waist circumference ([TRF] −2.52 ± 3.87 vs. [SC] 1.33 ± 3.39, *p* = 0.001), and BMI ([TRF] −0.59 ± 0.67 vs. [SC] 0.04 ± 0.71, *p* = 0.004, Table 2). The SC intervention caused a significant increase in HDL levels compared to TRF ([TRF] 0.003 ± 0.19 vs. [SC] 0.01 ± 0.21, *p* = 0.005, Table 2). The 12-week TRF intervention led to significant within-group improvements in weight, BMI, waist circumference, systolic blood pressure, diastolic blood pressure, ALT, and total tissue fat percentage, as well as a decrease in lean muscle mass (Table 3). The SC intervention led to an increase in waist circumference (*p* = 0.037) only. There were no significant within-group changes in LDL or total cholesterol, triglycerides, or heart rate over either intervention period (Table 3).

##### Metabolic and Inflammatory Biomarkers

Both the SC and TRF interventions led to increases in adiponectin concentrations over the course of the 12 weeks ([TRF] *p* ≤ 0.001, [SC] *p* ≤ 0.001), as well as decreases in leptin concentrations ([TRF] *p* = 0.033, [SC] *p* = < 0.001); however, there were no significant differences between the two interventions. While there were no significant differences between the two interventions, only the SC control was associated with an increase in insulin concentration (*p* = 0.033) and a decrease in HOMA-IR (*p* = 0.042). There were no significant between- or within-group differences in the inflammatory markers measured (TNFα, or FCP).

##### Diet, Hunger, and Activity

Total caloric intake did not change in either the TRF or SC groups. Additionally, there were no changes in the percentage of energy obtained from total fats, saturated fats, proteins, or carbohydrates throughout the study or between the groups. VAS scores for hunger, satisfaction, or happiness with diet did not change in either group over the 12 weeks (Table 5).

## 4. Discussion

NAFLD is highly prevalent and is associated with adverse long-term health outcomes. Hence, effective, easily administered therapies are needed. Although many lifestyle modifications involve the goal of losing weight through CR and exercise, there has been an increasing interest in TRF with and without CR as a treatment for obesity [12]. To our knowledge, this is the first randomized controlled trial comparing TRF to standard advice regarding diet and exercise in patients with NAFLD, although there has been increasing interest in its use in the management of overweight and obesity [12,20]. We explored this regimen as a lifestyle intervention for patients with NAFLD and found that the intermittent fasting intervention offered significant benefits compared to standard care, with improvements to steatosis, weight, BMI, and waist circumference. Interestingly, the fasting intervention also saw improvements in other key cardiometabolic measures, such as blood pressure, visceral fat, and total body fat, despite these changes not reaching significance compared to the standard care control. This paper adds to the growing literature on the effect of time-restricted eating on important metabolic and hepatic parameters [12,20].

IF is a more attractive lifestyle modification because it is simpler to restrict the time at which one eats rather than what is being eaten. The main strength of our study is that it accurately presents results for individuals with NAFLD typically seen in the outpatient setting. In addition, TRF advice is easy to follow, and the effects are easily measurable via point-of-care assessments, TE, and anthropometry, and these assessments can be conducted at little expense, with immediate benefits in terms of helping to guide decision making.

NAFLD treatment success is determined by a reduction in hepatic steatosis, inflammation, and fibrosis. We used the CAP function of FibroScan^®^ (M-Probe) to determine liver steatosis, as it measures the degree of ultrasound attenuation due to hepatic fat. CAP has been shown to correlate well with the amount of steatosis [16,18], and values > 215 dB/m are ≥ 90% sensitive for steatosis in ≥ 10% of hepatocytes [18]. When we analysed those with baseline CAP > 268 dB/m, there was a greater reduction in the TRF intervention. It is possible that CAP values below 268 dB/m are unreliable in identifying fatty liver or changes in steatosis over time. This suggests monitoring WC changes may be a simpler way to monitor the effectiveness of therapies targeted at central adiposity and liver steatosis. Despite the reductions in CAP shown and the strength of CAP as a non-invasive indicator of hepatic steatosis, true measurement via magnetic resonance imaging with proton density fat fraction would provide a stronger indicator of change.

Hepatic fibrosis is the main predictor of NAFLD progression [21]; however, liver biopsy is an invasive, costly, and undesirable way to monitor fibrosis progression or regression. The determination of LSM via TE is non-invasive and correlated with the histological fibrosis stage in NAFLD [22]. While neither intervention showed a decrease in LSM over the 12 weeks, this may be considered relatively unlikely in such a short time frame. Improvements in steatosis and NASH histology have been observed, with a 5–10% increase in weight loss [23]; however, the TRF group only achieved a 1.72 kg loss, making the improvements in CAP particularly noteworthy. Improvements in visceral fat are independently associated with improvements in liver fibrosis [24]. In our study, only the TRF group had significant improvements in these parameters over the duration of the intervention. TRF offered superior improvements in waist circumference (an important indicator of visceral fat) over SC, as well as providing a within-group improvement in visceral fat of 896 cm^2^. Similar results were not observed in the SC group. These measures of central adiposity and visceral fat also correlate with liver steatosis [25], which is also independently associated with liver fibrosis [26], providing a strong rationale for TRF as a primary intervention for NAFLD.

We considered caloric reduction as an explanation for the significant changes seen in the TRF groups; however, there was no change in total energy consumption from baseline or through the study in either the TRF or SC participants. On average, each group consumed 7500 kJ/day. The fact that the caloric intakes of the groups did not change, only the time during which calories were consumed can account for the changes associated with fasting, which may explain why the TRF individuals lost significant visceral/abdominal fat. A randomized trial by Koopman et al. showed that despite having the same high caloric intakes, individuals who ate more often rather than consuming three large meals without snacking had more liver steatosis and abdominal fat independent of weight gain [27]. This study suggests that spreading calories out over the day contributes to obesity and fatty liver. In our trial, participants in the TRF arm were instructed to eat for only 8 h each day, which limited snacking time between meals. Those in the SC arm were able to eat throughout the day. This difference may have contributed to the changes seen in visceral adiposity and liver stiffness.

There was no change in fasting insulin, glucose, cholesterol TNF-α, or ALT over the 12-week study period in either the TRF or SC groups, likely due to the normal baseline biochemical markers in most participants. However, significant increases in adiponectin and significant decreases in leptin were observed. In the liver, adiponectin increases insulin sensitivity and has anti-fibrotic and anti-inflammatory effects [28]. Increased adiponectin levels in NAFLD are associated with improved liver histology and lower cardiovascular risk [29,30]. Leptin is thought to have pro-inflammatory and pro-fibrotic effects in NAFLD [31]. Adiponectin increases and leptin decreases as fat mass decreases [28,32], and this may explain our findings, as we observed decreases in total fat mass in both the SC and TRF groups.

An association between weight loss and BP reduction has been well established through CR with or without exercise [33]. There was a > 4 mmHg reduction with the TRF intervention, a clinically relevant improvement, as a 3 mmHg reduction in SBP equates to an 8% reduction in stroke mortality and a 5% reduction in mortality due to coronary heart disease [34].

Throughout the study, neither caloric intake nor activity levels significantly changed in either group despite the dietary and exercise advice given to the SC group. Most TRF participants (88%) adhered to the 8 h TRF. The 12% lack of compliance (52/422 days) was mainly due to adding milk to morning coffee. Both diets were well accepted, with no change in hunger scores, happiness, or diet satisfaction.

Our study had a number of limitations. NAFLD was diagnosed via ultrasound, the most commonly used diagnostic tool for the detection of hepatic steatosis, with a reported sensitivity of 60–94% and a specificity of 66–95% [35]. We used FibroScan^®^ to determine liver stiffness and hepatic steatosis; however, these outcomes were not confirmed with liver histology. The small number of participants in each group did not allow for the determination of the differences between the SC and TRF groups. Finally, subjects’ dietary histories were self-reported, so true adherence rates to TRF may have been lower than reported.

At the time of this study, the standard-of-care dietary advice for fatty liver was general advice on reducing calories and fats. The guidelines published in 2021 are more detailed and recommend (but do not exclusively prescribe) the Mediterranean diet. Future studies should compare advice on the Mediterranean diet and TRF.

## 5. Conclusions

TRF and standard diet/exercise advice were both effective in decreasing body weight and improving leptin and adiponectin values. However, TRF may be superior in targeting steatosis and visceral adiposity, as demonstrated by decreased values for WC and visceral fat volume in the TRF group. These changes occurred without a reduction in calories during TRF, indicating that the timing of nutritional intake may be as important as the type of intake itself. These changes are clinically relevant and are novel findings in the context of short-duration lifestyle intervention studies. This pilot study provides sufficient evidence to support a larger randomized controlled trial in a well-defined NAFLD population.

## Figures and Tables

**Figure 1 nutrients-15-04870-f001:**
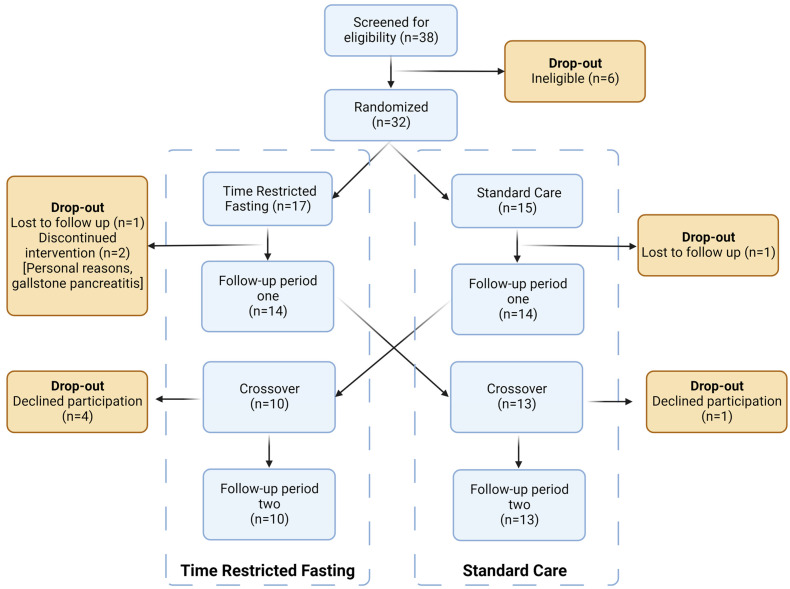
CONSORT patient flow diagram.

**Table 1 nutrients-15-04870-t001:** Baseline characteristics of participants by study group sequence (first intervention labelled).

	Time-Restricted Fasting (*n* = 17) Mean (SD)	Standard Care (*n* = 15) Mean (SD)
Age (years)	61.9 (8.7)	54.7 (13.0)
Sex (M/F) [*n*]	10/7	9/6
Ethnicity [*n*] Caucasian South Asian East Asian Hispanic		
	13	10
	2	3
	2	0
	0	2
Weight (kg)	83.2 (14.4)	82.7 (11.5)
Height (cm)	167.4 (10.1)	166.4 (8.7)
BMI (kg/m^2^)	29.5 (3.0)	29.9 (3.3)
WC (cm) Males Females	101.8 (10.2)	100.4 (8.0)
	105.5 (2.79)	100.9 (2.68)
	96.6 (3.90)	99.7 (3.56)
Metabolic syndrome † [*n* (%)]	8 (47%)	7 (47%)
CAP (dB/m)	256.07 (59.96)	256.23 (50.96)
Visceral Fat (cm^2^)	17,836.45 (4270.24)	16,115.91 (5565.39)
Baseline daily energy intake (kJ)	7747 (829.34)	7307 (1881.26)

†: Based on presence of 3/5 diagnostic criteria—central adiposity, elevated triglycerides, reduced high density lipoprotein, elevated blood pressure, and elevated plasma glucose

**Table 2 nutrients-15-04870-t002:** Paired analysis of outcome change by intervention. CAP: Controlled Attenuation Parameter, SBP: Systolic Blood Pressure, DBP Diastolic Blood Pressure, BMI: Body Mass Index, ALT: Alanine Transaminase, LDL: Low-Density Lipoprotein, HDL: High-Density Lipoprotein.

Measure	Time-Restricted Fasting Mean Change (SD)	Standard Care Mean Change (SD)	Between-Intervention Difference	*p* Value
Weight (kg)	−1.72 (2.01)	0.13 (2.13)	1.85	0.005
Waist Circumference (cm)	−2.52 (3.87)	1.33 (3.39)	3.85	0.001
BMI (kg/m^2^)	−0.59 (0.67)	0.04 (0.71)	0.63	0.004
Liver Stiffness (Kpa)	0.02 (5.94)	0.21 (2.38)	0.19	0.559
CAP (dB/m)	−9.96 (51.03)	20.46 (49.15)	30.42	0.038
DBP (mmHg)	−4.00 (7.96)	−2.56 (9.53)	1.44	0.492
SBP (mmHg)	−4.67 (9.81)	−4.89 (16.54)	0.31	0.972
Heart rate (bpm)	0.21 (8.23)	2.56 (11.16)	2.35	0.726
ALT (ug/mL)	−10.00 (15.71)	−3.48 (20.86)	6.52	0.094
FPG (mmol/L)	0.27 (3.02)	0.48 (2.31)	0.21	0.380
Insulin (mU/L)	−0.52 (4.33)	0.67 (5.14)	1.19	0.459
HOMA-IR	−0.17 (1.36)	0.47 (1.21)	0.64	0.090
Total cholesterol (mmol/L)	0.07 (0.80)	−0.11 (0.76)	0.18	0.621
Triglycerides (mmol/L)	−0.004 (0.43)	−0.18 (0.50)	0.18	0.434
LDL (mmol/L)	0.19 (1.00)	0.17 (1.25)	0.02	0.396
HDL (mmol/L)	0.003 (0.19)	0.01 (0.21)	0.01	0.008
Visceral fat (cm^2^)	896.45 (2773.55)	212.42 (4697.78)	684.03	0.702
Total tissue fat (%)	−0.82 (1.30)	−0.52 (1.75)	0.3	0.278
Total lean mass (g)	325.21 (1351.09)	−354.29 (1094.92)	679.5	0.510
Adiponectin (ug/mL)	3.58 (4.44)	3.40 (3.91)	0.18	0.622
Energy intake (KJ)	163.91 (1897.79)	91.02 (1514.31)	72.89	0.816

**Table 3 nutrients-15-04870-t003:** Within-group changes over the 12-week intervention period. CAP: Controlled Attenuation Parameter, SBP: Systolic Blood Pressure, DBP Diastolic Blood Pressure, BMI: Body Mass Index, ALT: Alanine Transaminase, LDL: Low-Density Lipoprotein, HDL: High-Density Lipoprotein.

Outcome		Baseline Mean (SD)	Follow-Up Mean (SD)	Change	*p* Value
Weight (kg)	*Time-Restricted Fasting*	80.61 (12.05)	78.89 (11.72)	1.72	<0.001
	*Standard Care*	81.27 (12.52)	81.389 (12.85)	−0.12	0.761
BMI (kg/m^2^)	*Time-Restricted Fasting*	29.09 (2.67)	28.51 (2.98)	0.58	<0.001
	*Standard Care*	28.88 (2.93)	28.92 (3.00)	−0.04	0.767
WC (cm)	*Time-Restricted Fasting*	101.25 (8.95)	98.73 (8.25)	2.52	0.004
	*Standard Care*	99.94 (8.42)	101.28 (9.58)	−1.34	0.037
Liver Stiffness (Kpa)	*Time-Restricted Fasting*	7.63 (5.06)	7.33 (4.60)	0.3	0.135
	*Standard Care*	6.67 (2.71)	6.89 (3.91)	−0.22	0.655
CAP (dB/m)	*Time-Restricted Fasting*	265.42 (55.12)	255.46 (48.03)	9.96	0.049
	*Standard Care*	256.12 (51.29)	272.56 (49.31)	−16.44	0.050
SBP (mmHg)	*Time-Restricted Fasting*	134.50 (14.84)	129.83 (12.18)	4.67	0.029
	*Standard Care*	132.37 (14.94)	127.48 (15.26)	4.89	0.137
DBP (mmHg)	*Time-Restricted Fasting*	83.292(8.78)	79.29 (7.43)	4.00	0.022
	*Standard Care*	82.111(10.83)	79.56 (8.38)	2.55	0.175
Heart Rate (bpm)	*Time-Restricted Fasting*	67.29 (13.03)	67.50 (14.40)	−0.21	0.90
	*Standard Care*	66.41 (10.66)	68.96 (11.44)	−2.55	0.245
ALT (µg/L)	*Time-Restricted Fasting*	49.29 (25.85)	39.29 (18.01)	10.00	0.005
	*Standard Care*	50.52 (29.73)	47.04 (20.01)	3.48	0.533
FPG (mmol/L)	*Time-Restricted Fasting*	5.68 (1.36)	5.70 (3.04)	−0.02	0.155
	*Standard Care*	5.53 (1.80)	6.11 (2.85)	−0.58	0.354
Insulin (mU/L)	*Time-Restricted Fasting*	9.18 (8.20)	8.23 (6.23)	0.95	0.502
	*Standard Care*	6.10 (5.13)	6.80 (5.04)	−0.7	0.033
HOMA-IR	*Time-Restricted Fasting*	6.37 (5.45)	6.19 (7.11)	0.18	0.910
	*Standard Care*	6.91 (6.25)	6.86 (7.26)	0.05	0.042
Total cholesterol (mmol/L)	*Time-Restricted Fasting*	5.14 (1.34)	5.21 (1.28)	−0.07	0.670
	*Standard Care*	5.27 (1.30)	5.16 (1.38)	0.11	0.457
Triglycerides (mmol/L)	*Time-Restricted Fasting*	1.45 (0.62)	1.44 (0.68)	0.01	0.96
	*Standard Care*	1.60 (0.73)	1.42 (0.67)	0.18	0.077
HDL (mmol/L)	*Time-Restricted Fasting*	1.33 (0.34)	1.35 (0.36)	−0.02	0.936
	*Standard Care*	1.30 (0.25)	1.32 (0.29)	−0.02	0.849
LDL (mmol/L)	*Time-Restricted Fasting*	3.20 (1.02)	3.23 (1.06)	−0.03	0.868
	*Standard Care*	3.24 (0.96)	3.22 (1.11)	0.02	0.573
Visceral fat (cm^2^)	*Time-Restricted Fasting*	16,925.26 (4217.92)	15,646.89 (4635.85)	1278.37	0.029
	*Standard Care*	15,952.73 (5158.97)	16,132.29 (5572.79)	−179.56	0.711
Total tissue fat (%)	*Time-Restricted Fasting*	37.15 (7.55)	36.33 (7.90)	0.82	0.005
	*Standard Care*	37.06 (7.46)	36.11 (7.27)	0.95	0.133
Total Lean muscle mass (g)	*Time-Restricted Fasting*	48,783.50 (9928.91)	48,143.21 (9496.70)	640.29	0.018
	*Standard Care*	49,034.19 (9052.07)	50,120.77 (9257.96)	−1086.58	0.099
Adiponectin (ug/mL)	*Time-Restricted Fasting*	16.70 (7.82)	20.28 (8.83)	−3.58	<0.001
	*Standard Care*	17.37 (8.11)	20.78 (9.12)	−3.41	<0.001
Leptin (ng/mL)	*Time-Restricted Fasting*	8.27 (4.04)	7.35 (4.34)	0.92	0.033
	*Standard Care*	7.21 (3.42)	5.79 (3.86)	1.42	<0.001
TNF-α (pg/mL)	*Time-Restricted Fasting*	2.25 (1.32)	1.82 (0.57)	0.43	0.173
	*Standard Care*	1.60 (1.02)	2.04 (1.21)	−0.44	0.320
FCP (µg/mL)	*Time-Restricted Fasting*	158.32 (195.31)	223.32 (219.34)	−65	0.126
	*Standard Care*	98.00 (154.69)	138.50 (164.11)	−40.5	0.296
Energy intake (KJ)	*Time-Restricted Fasting*	7932.54(1313.93)	7768.63(2075.11)	163.91	0.653
	*Standard Care*	7728.65 (1974.61)	8007.13 (1732.49)	−278.48	0.776

**Table 4 nutrients-15-04870-t004:** Outcome change between interventions in participants with significant baseline liver steatosis (CAP ≥ 268 dB/m). CAP: Controlled Attenuation Parameter; ALT: Alanine Transaminase.

Measure	Time-Restricted Fasting Mean Change (SD)	Standard Care Mean Change (SD)	Between-Intervention Difference	*p* Value
CAP (dB/m)	−26.00 (33.01)	15.10 (27.76)	41.1	0.035
Liver Stiffness (Kpa)	0.57 (9.21)	0.10 (3.35)	0.37	0.413
ALT (ug/mL)	−13.40 (17.49)	−2.80 (11.49)	10.6	0.212

**Table 5 nutrients-15-04870-t005:** Satiety and physical activity.

	Group	Week 0	Week 4	Week 8	Week 12	*p* Value
VAS score						
Hunger over last 24 h	TRF	4.4 (1.2)	4.0 (1.4)	3.7 (2.2)	4.8 (1.8)	NS
	SC	4.1 (2.3)	3.6 (1.7)	4.2 (2.2)	4.8 (2.0)	
Satisfaction over last 24 h	TRF	5.7 (1.8)	5.6 (1.7)	5.3 (1.7)	5.5 (1.8)	NS
	SC	7.0 (2.0)	6.7 (1.4)	6.1 (2.5)	6.3 (1.5)	
Happiness with diet over last 24 h	TRF	5.2 (1.3)	6.5 (1.8)	6.4 (1.9)	6.5 (2.1)	NS
	SC	5.7 (2.0)	6.2 (1.7)	5.7 (2.2)	6.0 (1.8)	
GPAQ						
Metabolic equivalent hours (kcal/kg) per week	TRF	45.4 (44.6)	53.5 (52.8)	54.2 (66.9)	66.5 (90.9)	NS
	SC	40.6 (53.3)	52.4 (53.5)	37.3 (37.3)	45.1 (43.2)	

## Data Availability

All data reported in this study are available from the corresponding authors upon reasonable request.

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
