# Peer review of "Time-Restricted Fasting Improves Liver Steatosis in Non-Alcoholic Fatty Liver Disease—A Single Blinded Crossover Trial"

_nutrients, 2023, doi:10.3390/nu15234870_

Round 1

Reviewer 1 Report

Comments and Suggestions for Authors

In ‘Time restricted fasting improves liver steatosis in non-alcoholic fatty liver disease – a single blinded cross over trial’, Feehan et al. show benefit of a 8h eating window/ 16h overnight fast, on reducing CAP, an index of NAFDL.

The intervention restricted eating to a window of 8h between noon to 8PM. Black tea and coffee were permitted. Outcome were liver fat by US/CAP and fibroscan.

Measures included: body composition, fecal calprotectin, fasting metabolic blood parameters, diet by 3day self-reported food diary, and hunger by VAS, GPAQ

Main finding is No differences were seen between assignments in liver stiffness but greater decrease of CAP during IF phase, and a trend to decrease ALt.

This is a nice pilot study.

Strengths: cross over design; follow up 24 weeks after end of intervention; signle fibroscan operator blinded to assignment.

VAT by CT single slice L4/L5- Single operator to analyze images.

The main limitations are: 12 weeks duration; power calculated for 58 participants, however 32 only recruited and 23 completers only.

A few issues were noted

Title and abstract refer to Time restricted fasting and intermittent fasting (IF). A bit confusing to have 2 different names. In addition, neither term exactly reflects the intervention which is a prolongation of the overnight fast duration to 16h resulting in a reduction of the eating window to 8h. Later in the manuscript, to add confusion ‘Time restricted feeding TRF’ is used. The IF term is also confusing. This was continuous prolonged overnight fast, not intermittent. Please use only one meaningful term to describe the intervention.

Study design:

Please give details Re control phase (SC) intervention. How long was the diet counseling session?

How many interactions between staff and participants in each assigned intervention (IF or SC)?

DEXA does not measure ‘muscle mass’. Please correct.

In table2, the ‘between group diff’ is misleading. It is the same group of individuals undergoing 2 diff interventions/ phases of the study. Perhaps use between intervention or phase differences?

Are changes with each intervention calculated from baseline (prior to any intervention) or from data point just before cross over?

What was the duration of the wash out period?

How was the order of assignment to IF vs SC tested in the analysis? For individuals assigned to IF first, did they maintain their weight loss when crossed over to SC?

Please present individual weight trajectories in a supplemental figure.

Body composition: fat mass is presented in % and lean mass in gr. Please add gr for fat mass too.

The duration of the overnight fast prior to the measurement of liver fat is an important factor. How was this controlled for and monitored in each group (IF and SC)? Was the time at which the fibroscan performed recorded? Please provide the duration of the overnight fast : mean, SD, range

How was adherence to the eating window measured?

The SC group did receive dietary information and no change in the diet composition pre and post SC phase were noted. While there was no nutrition info give to the IF phase, participants may have improved their diet composition. The diet diary may not have been sensitive enough to pick up small changes in diet composition in either phase of the study.

While weight did not change in SC phase, some parameters – usually linked to weight change- improve similarly in SC and IF.

How do you explain the change in leptin and adiponectin in both IF and SC phase of the study?

The SBP decrease by the same clinically significant magnitude in both phases of the study, while statistically significant in IF only.

HOMA IR improve in SC.

Please comment. Could the methods to measure food consumption and physical activity not be sensitive enough to pick up small changes?

Author Response

We thank the reviewer for their detailed comments and have incorporated all feedback into the manuscript. See the attached for a point by point explanation of changes.

Reviewer 2 Report

Comments and Suggestions for Authors

Dear Autors,

I have carefully reviewed the manuscript titled "Time restricted fasting improves liver steatosis in non-alcoholic fatty liver disease – a single blinded cross over trial " and appreciate the efforts made by the authors. I would like to provide some feedback and suggestions to improve the clarity and overall quality of the work.

Cut-off Values for Cap Score in the Methods Section: In the Methods section, it is essential to include a detailed description of the cut-off values used for the Cap Score measurement with the Fibroscan. This information is crucial for the readers to understand how the Cap Score was calculated and to ensure the reproducibility of the results. Please add this information to enhance the transparency and comprehensibility of your methodology.

Sample Size and Clinical Applicability: One of the significant concerns I have is the relatively small sample size used in this study. Given the importance of the Cap Score in clinical practice, it's essential to demonstrate that the results are not only statistically significant but also clinically relevant. To address this concern, I recommend discussing the study as a pilot study explicitly. Clarify that the study's small sample size limits the generalizability of the findings to a broader clinical population. By defining it as a pilot study, you can acknowledge the preliminary nature of the research and the need for further investigations with larger sample sizes.

Author Response

We thank the reviewer for their feedback. We have addressed their comments in the revised manuscript, and documented the changes in the attached document.
